# Register Always Matters: Analysis of LLM Pretraining Data Through the Lens of Language Variation

**Amanda Myntti, Erik Henriksson, Veronika Laippala & Sampo Pyysalo**
University of Turku
{amanda.a.myntti,erik.henriksson,mavela,spyysalo}@utu.fi

## Abstract

Pretraining data curation is a cornerstone in Large Language Model (LLM) development, leading to growing research on quality filtering of large web corpora. From statistical quality flags to LLM-based labelling systems, datasets are divided into categories, frequently reducing to a binary: those passing the filters are deemed as valuable examples, others are discarded as useless or detrimental. However, a more detailed understanding of the contribution of different kinds of texts to model performance is still largely lacking. In this article, we present the first study utilising *registers* or *genres*—a widely used standard in corpus linguistics to model linguistic variation—to curate pretraining datasets and investigate the effect of register on the performance of LLMs. We train small generative models with register classified data and evaluate them using standard benchmarks, and show that the register of pretraining data substantially affects model performance. We uncover surprising relationships between the pretraining material and the resulting models: using the *News* register results in subpar performance, and on the contrary, including the *Opinion* class, covering texts such as reviews and opinion blogs, is highly beneficial. While a model trained on the entire unfiltered dataset outperforms those trained on datasets limited to a single register, combining well-performing registers such as *How-to-Instructions*, *Informational Description*, and *Opinion* leads to major improvements. Furthermore, analysis of individual benchmark results reveals key differences in the strengths and drawbacks of specific register classes as pretraining data: *How-to-Instructions* excels at physical reasoning and sentence completion while barely crossing random baselines on world-knowledge benchmarks, while *Narrative* boosts performance on social interaction tasks but struggles with scientific questions. These findings show that register is an important explainer of model variation and can facilitate more deliberate and detailed future data selection practices.

## 1 Introduction

Recent work on scaling laws and the limits of human-generated training data available for Large Language Model (LLM) training have caused an uptick in research on pretraining data quality as opposed to quantity (Kaplan et al., 2020; Hoffmann et al., 2022; Muennighoff et al., 2023; Villalobos et al., 2024). While state-of-the-art models are still trained predominantly on large web corpora, there is increasing focus on filtering such data to remove potentially detrimental material. The selection of pretraining data often relies on simple heuristics or quality signals (Longpre et al., 2024; Ostendorff et al., 2024), sometimes together with LLM-created quality flags (Weber et al., 2024; Henriksson et al., 2025). However, language use varies along a number of dimensions other than the presumed dichotomy of positive or negative value for LLM training, ranging from persuasive, narrative, to informative texts. The effects of this variation on LLM capabilities remain unknown. In this paper, we approach pretraining data curation from a novel perspective: how do different registers (or genres) of training data affect LLM performance on commonly used benchmarks.

*Register studies* is a widely applied paradigm in corpus linguistics that examines language variation across different situations, from casual spoken conversations to persuasive and informational settings (Biber, 1988; 1995; Biber & Conrad, 2019). Registers are defined as situationally characterised text varieties, with typical classes encompassing news, reviews and song lyrics. Register has been shown to be one of the most important predictors of linguistic variation (Biber, 2012)—and, as put by the founder of register studies, Douglas Biber, "*register always matters*" (Biber, 2013).

Web register classification has long aimed to characterise different kinds of web texts in order to better utilise and understand the web as a corpus (Kilgarriff & Grefenstette, 2003; Baroni & Kilgarriff, 2006). However, early studies focused on pre-selected sets of registers presumed to be typical of the web, and thus lacked generalizability (Santini, 2007; Sharoff et al., 2010) and subsequently failed to capture the full range of linguistic variation found on the unrestricted web. Egbert et al. (2015) addressed this gap by creating a corpus representing the full spectrum of English web registers, with Laippala et al. (2023) presenting the first deep learning-based web register classifier trained on the data and targeting the entire unrestricted web. Other studies expanded this approach to multilingual settings (Repo et al., 2021; Rönnqvist et al., 2021; Kuzman et al., 2023a; Henriksson et al., 2024), showing that web registers are identifiable in web-scale data. These advances now enable the integration of corpus linguistics frameworks with LLM pretraining data curation—our study being to the best of our knowledge the first to bridge these areas, despite the well-documented role of registers in linguistic variation.

To understand the characteristics of web registers in the context of LLM pretraining, we train small generative models up to 100 billion tokens with web data with datasets curated to only contain texts from one register class. We evaluate these models using well-known benchmarks and analyse the results through average accuracy and performance on individual benchmarks, revealing how each register impacts model capabilities. Our results show that register has a substantial impact on model performance. Additionally, we find that training on a combination of register-specific datasets that displayed strengths in different benchmarks can lead to major improvements: registers like *Opinion* boosts model performance, while incorporating examples from register *Interactive Discussion* or *Spoken* at the expense of *How-to-Instructions*, *Informational Description* and *Opinion* leads to performance drops. With these findings, our study aims to open up discussion about linguistic selection of training data, and to show that register classification holds potential as a tool for analysing pretraining data.

## 2 Related work

Ablation studies, where small generative models are trained with different parts of a large dataset and evaluated on downstream tasks, are a well-established method for evaluating dataset quality and optimising data curation methods. Gao et al. (2020) validated the quality of their Pile datasets by evaluating models trained on the Pile, CommonCrawl, and CC-100. Longpre et al. (2024) pretrain small models on data filtered by different qualities (data age, data domain, data toxicity) and analyse the effects of this curation via model evaluation. Similarly, Sachdeva et al. (2024) use an ablation framework to find that diversity and coverage of different topics in the pretraining data are crucial to models that succeed in multiple benchmarks. Likewise, Burchell et al. (2025) released a multilingual dataset and performed ablation studies to measure the quality of different versions of their datasets. Henriksson et al. (2025) use an LLM to line-by-line classify training documents to optimise dataset composition and evaluate with a downstream task.

The methodology of our approach most closely resembles that of the FineWeb ablation studies (Penedo et al., 2024a). The FineWeb dataset was created by conducting analyses of tens of heuristic filters to select the data curation methods that result in the best pretraining data with respect to the downstream performance of the trained model. Currently, datasets produced by the FineWeb pipeline are available for over a thousand languages (Penedo et al., 2024b).

Current leading work on dataset composition optimisation includes Su et al. (2024) and Li et al. (2024). Lee et al. (2022) established deduplication as a standard procedure in pretraining data selection pipelines. Other common approaches for selecting pretraining data from web sources include URL-based filtering (e.g. Penedo et al. (2023)), text statistics such as word count, repetition, or presence of blacklisted words (e.g. Raffel et al. (2020)), filtering based on perplexity (e.g. Muennighoff et al. (2023),Weber et al. (2024)), and selection based on similarity against a selected good-quality corpus (e.g. Brown et al. (2020), Tirumala et al. (2023)). According to a survey by Albalak et al. (2024), it remains an open question whether quality filtering data improves model performance, and in which cases: Dodge et al. (2021) show that url-based pretraining data filtering leads to bias in the resulting data, while reference corpora used for similarity selection can be biased in language, topics, or demographics (Rae et al., 2022; Gururangan et al., 2022).

While register studies is a widely applied standard to analyse language variation in corpus linguistics, the term *genre* is more often used in the NLP field (Sharoff et al., 2010; Petrenz & Webber, 2011; Sharoff, 2020; Kuzman et al., 2023b). However, both approaches target text categories such as opinionated, informational, and lyrical.

Register studies has a long tradition in mapping the relationship between typical linguistic features associated with texts and different situational contexts where language is used, in both online and offline settings (e.g., Biber (1995); Biber & Egbert (2018); Ansarifar et al. (2025)). Biber (2012) even shows that register is one of the most important predictors of linguistic variation. Similarly, web register (or genre) classification is a widely applied approach to classify web content (Kilgarriff & Grefenstette, 2003; Santini, 2007; Petrenz & Webber, 2011; Myntti et al., 2024).

Since the release of the first web register corpus covering the full unrestricted web (Egbert et al., 2015), automatic register classification using machine learning has been successfully applied to the unrestricted web, achieving nearly human-level performance (Laippala et al., 2023; Kuzman et al., 2023b). Henriksson et al. (2024) also show that register characteristics transcend language boundaries, allowing classification models to identify registers even in languages that were not present in the training data (zero-shot languages). These advances enable the current study to select pretraining data by register. Our approach also aligns with calls for greater transparency in foundation models (Bommasani et al., 2023), as it provides structured metadata about the linguistic composition of training data, helping researchers and users to understand what types of texts a model has been exposed to during training.

## 3 Methods

### 3.1 Data

We use HPLT version 2.0[1] (Burchell et al., 2025) as our source data. This collection includes datasets for 193 languages and additional parallel datasets for 50 languages paired with English. HPLT v2 datasets have been processed from a combination of Internet Archive and Common Crawl using Trafilatura (Barbaresi, 2021) with language identification performed using OpenLID (Burchell et al., 2023). Two versions of each dataset are provided, *deduplicated*, with MinHash deduplication (Broder et al., 1998), and *cleaned*, with additional cleaning heuristics applied. An evaluation by Burchell et al. (2025) showed the cleaned English dataset to be of high quality, with models trained on the data achieving performance comparable to ones trained on the FineWeb dataset (Penedo et al., 2024a). In this study, we nevertheless chose to use the deduplicated English dataset, which yields slightly worse performance in LLM pretraining. This choice allows us to examine the effect of register with minimal interference from cleaning procedures that might disproportionately affect certain registers and thus bias our analysis. This decision is also motivated by the uneven distribution of register labels in web data; with fewer cleaning steps, we retain more data, which is especially important for the less frequent register classes.

---

[1]Available at `https://hplt-project.org/datasets/v2.0`

| Individual registers | Description | Available |
|---|---|---|
| *How-to-Instructions* (HI) | Recipes, instructions | 100 B |
| *Interactive Discussion* (ID) | Discussion forums | 314 B |
| *Informational Description* (IN) | Wikis, information sites | 695 B |
| *Informational Persuasion* (IP) | Advertisements, commercial sites | 421 B |
| *Lyrical* (LY) | Song lyrics, poems | 20 B |
| *Machine Translation* (MT) | Machine translation | 306 B |
| *Narrative* (NA) | News, blogs | 545 B |
| *Opinion* (OP) | Opinionated texts, religious sermons | 416 B |
| *Spoken* (SP) | Interviews, speeches | 32 B |
| *Instructive-Informational* (HI-IN) | Hybrid; Documents predicted as both HI and IN | 70B |
| *News* (ne) | Subregister of NA; news | 404 B |
| *Description* (dtp) | Subregister of IN; decription of a thing or a person | 781 B |
| **Combinations** | | |
| *HI-IN-HI-dtp* | 1/3rd sampled from HI-IN, HI, dtp | – |
| *HI-IN-HI-dtp-OP* | 1/4th sampled from HI-IN, HI, dtp, OP | – |
| *HI-IN-HI-dtp-OP-NA* | 1/5th sampled from HI-IN, HI, dtp, OP, NA | – |
| *HI-IN-HI-dtp-OP-NA-ID* | 1/6th sampled from HI-IN, HI, dtp, OP, NA, ID | – |
| *HI-IN-HI-dtp-OP-NA-ID-SP* | 1/7th sampled from HI-IN, HI, dtp, OP, NA, ID, SP | – |
| **Baselines** | | |
| HPLT v2 deduplicated | Burchell et al. (2025) | – |
| FineWeb | Penedo et al. (2024a) | – |

Table 1: Datasets used in our experiments. Each dataset corresponds to one trained model. The *Individual registers* section describes the main register labels and hybrid & subregisters. The *Combinations* section contains combined register classes, and *Baselines* lists the baseline datasets we use in our experiments. The *Available* column contains the approximate token counts available in the HPLT v2 deduplicated corpus.

The HPLT v2 dataset includes predictions of register classes as probabilities for each document, generated using the multi-label register classifier by Henriksson et al. (2024). This classifier was fine-tuned from XLM-RoBERTa-Large (Conneau et al., 2020) using the multilingual CORE corpus introduced in the same study. The CORE register scheme is hierarchical, with 9 main registers divided into 25 subregisters that further describe the content of each document. Documents can have none, one, or multiple main registers (referred to as *hybrids*), along with any associated subregister labels. The full register scheme can be found in Henriksson et al. (2024), and example documents are given in Appendix A.

Although the register labels are available for multiple languages, we limit our analysis to English due to the availability of varied and well-established benchmarks. For our experiments, we assign register labels to documents from the given probabilities using a classification threshold of 0.4, optimised for English. From the register classes, we select all main registers, 2 subregisters, and one hybrid for our experiments. We included the subregisters and hybrids for specific reasons: the subregisters *News* and *Description* are among the largest subregister classes, each with over 100 billion tokens available—sufficient data to train models and draw meaningful comparisons with their parent registers *Narrative* and *Informational Description*. Moreover, there has been particular interest in using news articles (e.g. de la Rosa et al. (2025)) and web encyclopedias (i.a. Wikipedia) to train LLMs, which correspond to classes *News* and *Description* in the register scheme. The hybrid class *Instructive-Informational* was included to study the differences between individual registers and hybrids, based on preliminary results showing that register-specific models trained on these classes perform well overall but differ across individual benchmark tasks. The selected register classes and their abbreviations are described with examples in Table 1. Subregister classes not selected for our experiments are present in the data; however, they merged to their main-level class.

Using these labels, we sample 100 billion tokens per register, using all available data for registers with fewer than 100 billion tokens. We limit our analysis to documents over 200

characters and remove documents with extremely high token counts (corresponding to over 300,000 words[2]) due to hardware constraints. These excluded documents represent less than 0.3% of the total data and primarily belong to the *Informational Description* class.

We also experiment with training on combinations of certain registers, which are listed in Table 1 under "Combinations". These combination datasets are created by sampling equally from individual register datasets: for example, in the combination dataset *HI-IN-HI-dtp*, 1/3rd of the training samples are from *Instructive-Informational* (HI-IN), 1/3rd from *How-to-Instructions* (HI), and 1/3rd from *Description* (dtp), with our dataset creation step ensuring no duplicates are present in the combined data. See Section 4 for further information and motivation for these datasets.

## 3.2 Model training

We replicate the training setup of Penedo et al. (2024a), using the same model architecture (Llama), model size (1.71 billion parameters), and other training parameters. In total, the models are trained up to 100 billion tokens. See Appendix B for further information on the training setup. As a baseline, we use a model trained on a random sample of the HPLT v2 deduplicated data. This allows us to compare the effects of using a single register as opposed to using data with the natural distribution of registers in the source dataset. Finally, as our setup mirrors that of Penedo et al. (2024a), we also include results for a model trained on a sample of the FineWeb dataset as a further point of comparison and as a reference for interpreting our findings.

## 3.3 Evaluation

To maintain comparability with previous work, we also follow Penedo et al. (2024a) in our evaluation. Specifically, we use the LightEval (Fourrier et al., 2023) evaluation harness with the following benchmarks in a zero-shot setting for all tasks:

- **HellaSwag** (Zellers et al., 2019) contains logical sentence completion tasks in the form of presenting a paragraph and an incomplete sentence with alternative sentence endings.
- **WinoGrande** (Sakaguchi et al., 2021) evaluates commonsense reasoning in a binary classification setting using pronoun resolution problems.
- **PIQA** (Bisk et al., 2020) focuses on physical commonsense reasoning, with examples involving manipulation of physical objects.
- **SIQA** (Sap et al., 2019) contains tasks on reasoning about social situations.
- **OpenBookQA** (Mihaylov et al., 2018) focuses on tasks requiring multi-step reasoning with additional context given.
- **ARC Easy and ARC Challenge** (Clark et al., 2018) are multiple choice benchmarks with science exam questions.
- **CommonsenseQA** (Talmor et al., 2019) consists of multiple-choice questions requiring prior world knowledge.
- **MMLU** (Hendrycks et al., 2021a;b) is a multiple-choice question answering dataset covering 57 distinct categories, such as US politics and anatomy.

These benchmarks were chosen by Penedo et al. (2024a) due to their compatibility with small model size, showing reliable, low-variance and monotonic signals even with limited training data, which is crucial for our experiments. For MMLU, we consider all 57 categories as one task and average our results accordingly. This ensures that models reaching high accuracy on MMLU but mediocre results on other benchmarks do not dominate the average results. See Appendix D for further discussion.

The selected benchmarks reflect the most common present-day expectations placed on LLMs: fluency, scientific and real-world knowledge, problem solving, and reasoning. However, this selection inevitably disadvantages certain register models. For example, models trained on the *Lyrical* register are likely to show poor performance, as none of the benchmarks

---

[2]Defined as white-space and punctuation separated segments

measure poetic capabilities that would be expected from a model trained on lyrical data. This does not mean that lyrical capabilities are unwanted or unnecessary for LLMs, but rather highlights the importance of benchmark diversity and the significance of selecting benchmarks that measure abilities relevant to specific use cases. In our case, although the benchmark selection disadvantages some registers and their expected performance more than others, using the full range of registers covering all kinds of web texts yields a complete picture of registers' effects on LLM pretraining.

## 4   Experiments

Our experiments are divided into two parts: First, we examine the effects of register on model performance by training models for the 9 main-level registers, one hybrid class (*Instructive-Informational*), and two subregister classes (*News* and *Description*), presented in Table 1 under "Individual registers". These experiments are meant to highlight the differences of registers as pretraining data.

Second, we investigate the effect of further mixing registers by selecting top-performing registers from our first experiment. This allows us to observe how adding or removing specific registers affects overall model performance and whether the strengths of individual registers are preserved in combined models. The selected top-scoring register combinations are presented in Table 1 under "Combinations". When sampling data for register combinations, we ensure that no duplicate documents are included, even when combining the hybrid class *Instructive-Informational* (HI-IN) with the register *How-to-Instructions* (HI). We chose to use the subregister *Description* (dtp) instead of the main-level register *Informational Description* (IN) because *Informational Description* showed signs of performance degradation in later training steps, and as noted in Section 3.1, we had to exclude some documents from the *Informational Description* dataset. To maintain comparability, we train all models up to 100 billion tokens, although the larger amount of combined data would allow for longer training. Our aim is not to identify optimal register combinations, but rather to explore how register-based data curation might influence model performance on different tasks.

Some of our datasets contain fewer than 100 billion tokens, which necessitates repeating data during training. This raises questions about the validity of our results, as Hoffmann et al. (2022) argue that repeating data can lead to performance degradation. However, Taylor et al. (2022) report validation loss continues to improve up to 4 epochs for a 120 billion parameter model, and Muennighoff et al. (2023) find that for the GPT-2 (Radford et al., 2019) architecture with 2-8 billion parameters, multi-epoch training has negligible effects when limited to 4 or fewer epochs. Xue et al. (2023) further demonstrate that smaller models are less susceptible to the negative effects of repeated training examples in T5-architecture models (Raffel et al., 2020). Therefore, we are reasonably confident in the results from models trained on *Instructive-Informational* (1.4 epochs) and *Spoken* (3.1 epochs). The model trained on our smallest dataset, *Lyrical* (5 epochs), performs poorly even within the first epochs, suggesting its low performance is due to register characteristics and benchmark selection rather than data repetition. None of the combination models exceed 1 epoch of training, since we sample the 100 billion tokens from combinations of multiple datasets, together exceeding 100 billion tokens.

We additionally compare our register scheme to three other quality classifiers, FineWeb-edu classifier (Penedo et al., 2024a), the DCLM quality classifier (Li et al., 2024), and NVIDIA's NemoCurator Quality Classifier DeBERTa (He et al., 2023). These experiments and their results are presented in Appendix C.

## 5   Results

### 5.1   Average performance

The results for individual register models, averaged over all benchmarks, are presented in Figure 1 and Table 2, and detailed in Appendix D. The evaluation shows striking differences in accuracy between the models, demonstrating that the register of the pretraining data

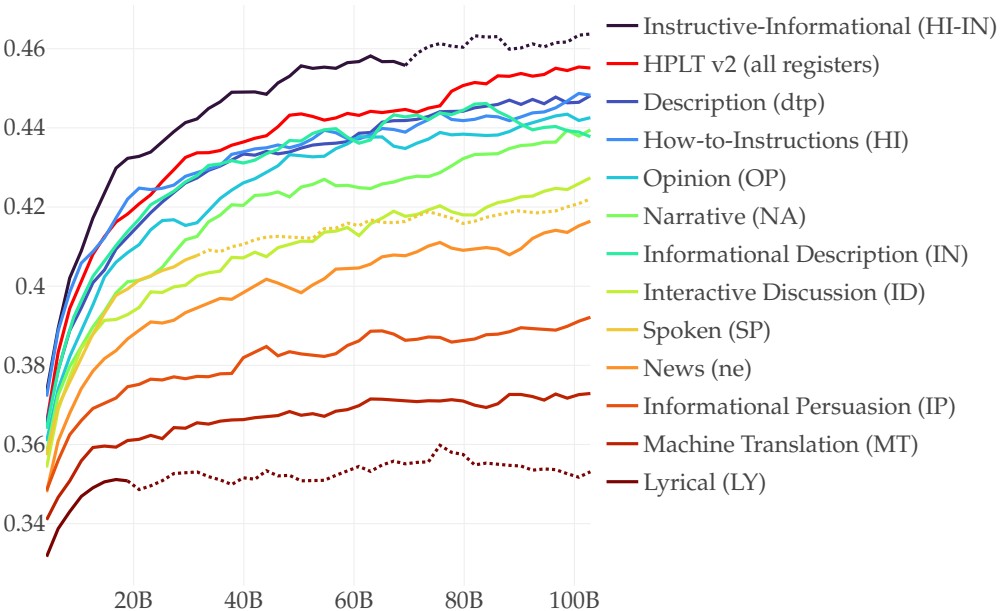

Figure 1: Individual register and HPLT v2 (deduplicated) model accuracies as a function of the number of training tokens. Rolling average over 6 billion tokens (3 adjacent checkpoints) applied for ease of reading overlapping lines, see Table 2 for numerical results for the final checkpoint. Dotted lines indicate training continuing over one epoch, legend in order of last checkpoint performance.

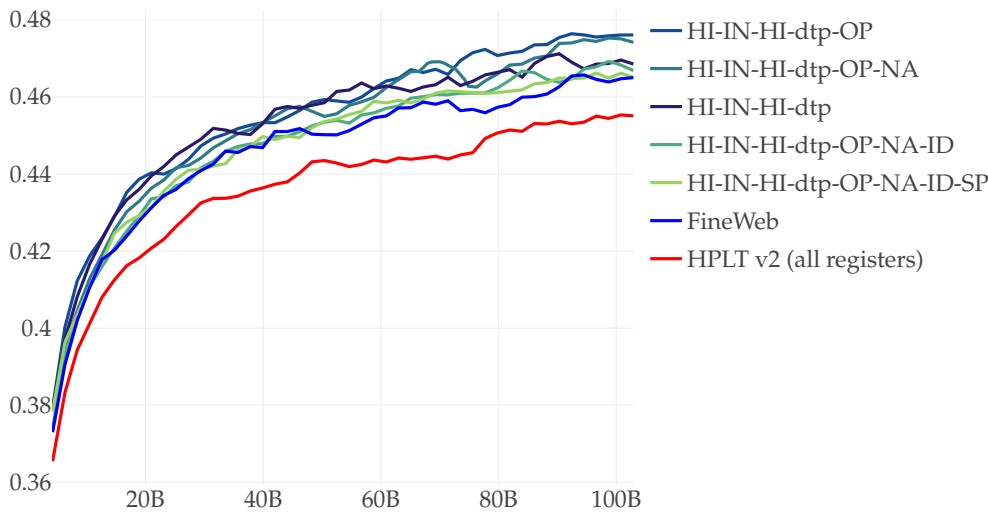

Figure 2: Combination model and baseline accuracies as a function of the number of training tokens. Rolling average over 6 billion tokens (3 adjacent checkpoints) applied for ease of reading overlapping lines, see Table 2 for numerical results for the final checkpoint. Legend in order of last checkpoint performance.

| Pretraining dataset | Accuracy | Pretraining dataset | Accuracy |
|---|---|---|---|
| *HI-IN-HI-dtp-OP* | 0.475 | *Opinion* (OP) | 0.447 |
| *HI-IN-HI-dtp-OP-NA* | 0.472 | *Narrative* (NA) | 0.441 |
| *HI-IN-HI-dtp* | 0.466 | *Informational Description* (IN) | 0.437 |
| FineWeb | 0.466 | *Interactive Discussion* (ID) | 0.431 |
| *HI-IN-HI-dtp-OP-NA-ID-SP* | 0.465 | *Spoken* (SP) | 0.422 |
| *Instructive-Informational* (HI-IN) | 0.465 | *News* (ne) | 0.418 |
| *HI-IN-HI-dtp-OP-NA-ID* | 0.464 | *Informational Persuasion* (IP) | 0.393 |
| HPLT v2 (deduplicated, all registers) | 0.457 | *Machine Translation* (MT) | 0.374 |
| *Description* (dtp) | 0.452 | *Lyrical* (LY) | 0.358 |
| *How-to-Instructions* (HI) | 0.447 | | |

Table 2: Results by the last checkpoint performance in numerical format. Note that the values differ slightly from the Figures 1 and 2 due averaging over adjacent checkpoints in the figures.

has a substantial effect on model performance. We find that the model trained on the full HPLT v2 data outperforms models trained on individual registers, demonstrating that a mix of registers is important for LLM training. The performance ranking of individual registers generally aligns with intuitions about their content and what is typically considered high-quality pretraining data: registers like *Lyrical* (LY), *Machine Translation* (MT), and *Informational Persuasion* (IP) yield worse-performing models, while models trained on registers containing informational content and instructions, such as *Description* (dtp) and *How-to-Instructions* (HI), perform the best. Surprisingly, *Opinion* (OP) yields the 4th best performing model. Both *Informational Description* (IN) and *Narrative* (NA) models perform worse than *How-to-Instructions*, *Description* and *Opinion*. Following these, with a noticeable performance drop, are the models trained on *Interactive Discussion* (ID), *Spoken* (SP) and remarkably, the subregister *News* (ne), consisting of news articles, which performs among the worst.

As opposed to the main register models, the hybrid class model *Instructive-Informational* (HI-IN) outperforms the model trained on the full HPLT v2 data. This shows that hybrid documents do not simply result in averaged performance between the two hybridised registers, but can show better performance than either alone. This finding motivated our second experiment, investigating register combinations. The subregisters *News* (ne) (subregister of *Narrative*) and *Description* (dtp) (subregister of *Informational Description*) show divergent patterns when compared to their parent registers: *News* drastically underperforms its main-level register *Narrative*, while *Description* achieves higher accuracy than *Informational Description*. We analyse differences between main-subregister pairs in more detail in the following section.

As the results show, pretraining a model with data from only one register class does not provide improvements compared to training with the full dataset. This aligns with the established notion that pretraining data should have variability to yield capable models. The results for our combination model experiment are shown in Figure 2, Table 2, and again with more detail in Appendix D. The results clearly support the above idea: all combination models outperform the individual register models and the baseline HPLT v2 model. Additionally, the best models in our study (*HI-IN-HI-dtp-OP* and *HI-IN-HI-dtp-OP-NA*) reach higher average accuracy than the models trained on the HPLT v2 and FineWeb datasets. *HI-IN-HI-dtp* model shows much worse performance than *HI-IN-HI-dtp-OP*, which further includes *Opinion* (OP). This suggests that opinionated texts provide valuable pretraining data and are key to high model performance, at least as measured by the selected benchmarks. Further adding the *Narrative* (NA) dataset, as seen in the *HI-IN-HI-dtp-OP-NA* model, slightly worsens performance, though the difference is small. Significant drops occur when including *Interactive Discussion* (ID) and *Spoken* (SP) in models *HI-IN-HI-dtp-OP-NA-ID* and *HI-IN-HI-dtp-OP-NA-ID-SP*, indicating either that these registers contain qualities that harm the abilities measured on these benchmarks, or that they do not provide benefits that outweigh the reduced proportion of *Instructive-Informational* (HI-IN), *Description* (dtp), and *Opinion* (OP) in the training mix.

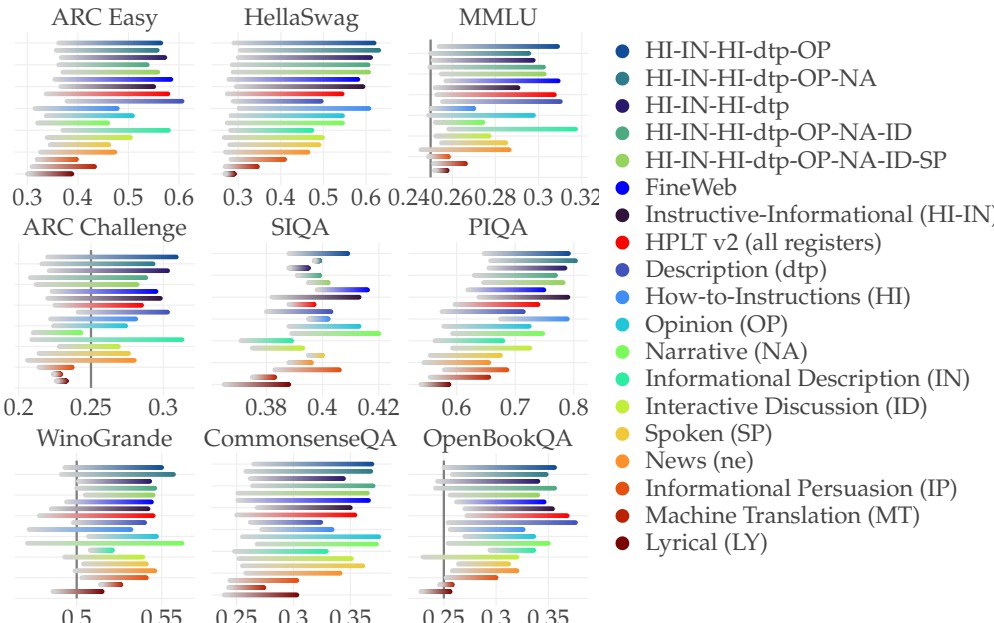

Figure 3: Change of accuracy from first to final checkpoint on individual benchmarks shown as a range, with grey indicating the first checkpoint and colours indicating the last checkpoint. The random-guess threshold is shown as a grey vertical line in cases where at least one model falls below it. Bars and legend shown in order of average accuracy.

## 5.2 Per task

The results for each benchmark separately are shown in Figure 3 and detailed further in Appendix E. The differences between benchmarks are remarkable, both in the performance ranking of models and in the way accuracy trends with respect to the number of training tokens. This shows that registers have strengths in different tasks. Most benchmarks display the monotonicity condition that influenced the selection of Penedo et al. (2024a), as they tend towards increased accuracy with more training tokens. A clear outlier is the SIQA benchmark, where increases in accuracy are very small, with the largest increase seen in *Narrative* (NA) and *Instructive-Informational* (HI-IN) models, both increasing approximately by 0.03 during the training.

Generally, the trends observed in Section 5.1 hold. The models with the lowest average accuracy—*Lyrical* (LY), *Machine Translation* (MT) and *Informational Persuasion* (IP)—perform poorly on all benchmarks, and, notably, worse than a random guess on ARC Challenge. The model for subregister *News* (ne), despite weak average performance, improves on the MMLU and WinoGrande benchmarks, while *Spoken* (SP) enhances results on CommonsenseQA. The differences between benchmarks become more pronounced with models trained on *Narrative* (NA), *Informational Description* (IN), *Opinion* (OP), *How-to-Instructions* (HI), and *Interactive Discussion* (ID), which display great variation in ranking across tasks, highlighting their strengths. The *Narrative* model performs well on SIQA and WinoGrande but barely surpasses the random baseline on ARC Challenge. *Informational Description* trumps other models on MMLU and both ARC benchmarks, but ranks low to mediocre elsewhere. The model trained on the *How-to-Instructions* dataset outperforms almost all other models on PIQA and HellaSwag. This might be because PIQA measures reasoning in physical tasks, which aligns with the instructions related to physics included in *How-to-Instructions*. On the other hand, *How-to-Instructions* ranks among the worst on MMLU and CommonsenseQA. The *Opinion* model excels in WinoGrande, SIQA and CommonsenseQA. This signal also appears in the combination models' performance on CommonsenseQA: including *Opinion* in the training data greatly increases accuracy on this benchmark, unlike the *HI-IN-HI-dtp* model, which excludes *Opinion*.

The hybrid *Instructive-Informational* (HI-IN) model generally performs between individual *How-to-Instructions* (HI) and *Informational Description* (IN) models, and is notably closer in performance to whichever yields better accuracy. Exceptions to this pattern are Open-BookQA, CommonsenseQA, and WinoGrande, where the *Instructive-Informational* model shows better accuracy than either *How-to-Instructions* or *Informational Description* separately. Specifically in the case of WinoGrande, this is surprising, as both *How-to-Instructions* and *Informational Description* struggle with this benchmark. From this result, we conclude that hybrids may contain qualities not found in individual registers and require further study. The *News* (ne) model trails behind *Narrative* (NA), except on the MMLU and ARC Challenge benchmarks. This is likely due to news articles covering topics related to MMLU tasks, such as politics, business, science, and the environment. Similarly, the other main-subregister pair, *Description* (dtp) and *Informational Description* (IN), shows differences in performance across benchmarks, with the *Description*-model outshining *Informational Description* in ARC Easy, HellaSwag, PIQA, WinoGrande, and OpenbookQA. *Informational Description*, on the other hand, excels in ARC Challenge and MMLU. Specifically for the WinoGrande benchmark, the performance difference might be explained by the *Description* register containing more texts focused on things and people, thus resulting in better pronoun-resolution abilities.

The combination models lead in performance on most benchmarks. On MMLU specifically, we observe a huge drop between the *HI-IN-HI-dtp-OP* model and others: including *Opinion* in the pretraining data increases performance, while removing *Opinion* or further adding *Narrative* greatly disrupts model performance on this benchmark. Notably, the models *HI-IN-HI-dtp-OP* and *HI-IN-HI-dtp-OP-NA* are trained with registers that individually perform well on a wide range of benchmarks: *Instructive-Informational* on HellaSwag and ARC Easy, *How-to-Instructions* on Hellaswag and PIQA, *Description* on ARC Easy, ARC Challenge, and OpenBookQA, *Opinion* on CommonsenseQA, and *Narrative* on SIQA and WinoGrande.

## 6 Conclusion

In this paper, we presented the first study investigating the effect of linguistic register on the performance of small generative LLMs. Our findings show that register is an important explainer of LLM performance, and we were able to show surprising relationships between the register of pretraining data and model accuracy on different benchmarks. While using the whole dataset yielded better performance than limiting the pretraining data to any individual register class, we found specific combinations of registers that outperform the full uncurated dataset. These findings can be used to increase pretraining data transparency and to optimise data selection methods, potentially addressing specific performance gaps by incorporating appropriate registers.

## Limitations and future work

As a first study in the intersection of two broad lines of research – the linguistic registers and LLM training – our work leaves a number of questions incompletely answered. Our experiments are conducted in English only. While register classification has shown multilingual prowess, our results might not generalise to other languages, and we leave multilingual analysis for future work. The benchmarks used, though standard in studies of this type, do not fully measure all model capabilities, as discussed regarding the *Lyrical* class. Evaluating with more diverse benchmarks would provide more comprehensive results and might reveal strengths and drawbacks we were unable to uncover.

Bias from data curation methods is known to propagate to trained models (Rae et al., 2022; Gururangan et al., 2022), and register classification is not exempt from this issue. To mitigate bias, we used the *deduplicated* version of the HPLT v2 data. Although this version has been processed with Trafilatura, Penedo et al. (2024a) note that Trafilatura can leave "undesirable" content in the dataset, potentially affecting some registers more than others. The possibility of benchmark contamination also exists; register classification could systematically assign leaked benchmark material to specific classes, potentially explaining some performance differences.

## Acknowledgments

This project has received funding from the *Finnish Doctoral Program Network in Artificial Intelligence, AI-DOC* under decision number VN/3137/2024-OKM-6, the European Union's *Horizon Europe research and innovation programme* under grant agreement No. 101070350, the *Digital Europe Programme* under grant agreement No. 101195233, and the Research Council of Finland under grant No. 362459. The contents of this publication are the sole responsibility of its authors and do not necessarily reflect the opinion of the European Union. We wish to acknowledge CSC - IT Center for Science, Finland, for computational resources.

## Ethics Statement

Training large language models is resource-demanding. This fact guided our decisions: we used small model sizes and trained only up to the point where clear conclusions can be drawn. In total, we used approximately 11,000 GPU hours per model. See Appendix B for calculation of the total computational cost. To further mitigate the environmental impact of training, we used LUMI supercomputer, which is among the most environmentally friendly supercomputers in the world[3]. The models trained for this study are planned to be used in future research.

Releasing our models publicly for reproducibility opens up the possibility of our models being used in contexts other than research. Although we do not evaluate the possible toxicity of the generations they produce, the model sizes are small, and they are trained only up to 100 billion tokens, which makes their generative capabilities weak and thus likely unsuitable for applications outside the scope of this study.

## Reproducibility statement

We have prioritised reproducibility in our work and make all artefacts created as part of this study, including code, model training configurations, and the trained model checkpoints, available on `https://huggingface.co/TurkuNLP/register-ablations` under open licenses. Our work only makes use of data openly available from `https://hplt-project.org/datasets/v2.0`, and our model training setup is detailed in Appendix B.

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

## A  Examples of texts in register classes

Below we show examples of texts classified as selected registers. These texts were chosen randomly but with a focus towards brevity (with some texts additionally truncated), general topics, and non-toxic language use. We also omit new-line characters.

*How-to-Instructions* (HI)

> Home & DIY How To Cut Your Own Hair At Home With countless non-essential businesses closing due to the government-implemented lock-downs, many people have been forced to be creative and resourceful in order to get by. Case in point, not everyone of us are lucky enough to be quarantined at home with a professional hairdresser. So we do the best we can – which often means cutting our own hair. [...]

*Interactive Discussion* (ID)

> I tried last night to compile a standalone which I've compiled approximately 1, 000, 000 times before, and have been met with a long series of very lengthy crash reports. As far as I can tell, something in the compiler is trying to install every framework on my computer, and I sometimes wind up with 1 gig+ .mxf files. Naturally, little dinky tests compile fine, and I can compile all of the modules that make up my patch, but not the whole thing at once. I did a super- clean Max reinstall, no third-party objects aside a couple of Lobjects which are up to date, and I'm stumped. The only difference I can think of since I compiled this a few weeks ago is an upgrade to 4.5. Anyone have any light to throw on this? Im happy to send crashlogs, but they're big. 10.5.2, Max 4.6, Jitter 1.6.3 thanks, M

*Machine Translation* (MT)

> Incredible Interior Design Sketches Good Looking Interior Designer Remarkable Interior Design Exceptional Interior Design Sketches Cozy Interior Design Sketches Chic Interior Design Sketches Cool Interior Design Sketches

*News* (ne)

> Wal-Mart has signed a storm water settlement agreement with the Connecticut Department of Environmental Protection concerning alleged violations at 20 Wal-Mart stores and two SAM'S CLUB locations in the state. Under the agreement, Wal-Mart will pay $600,000 in civil penalties for violations alleged to have taken place between 1996 and 2003. Wal-Mart also will contribute $550,000 to two different supplemental environmental projects– $500,000 to assist municipalities in addressing storm water issues, and $50,000 for environmental projects in the Connecticut River Watershed. [...]

*Opinion* (OP)

> Once again a very nice stop for the night. Staff has been very well trained, all are friendly and helpful with one exception/the young man who checked us in. Breakfast was so appreciated after 12 nights at a Holiday Inn Express which was ok but does not have your choices. Lynne does an especially good job/appreciated all her help. They... More

## B  Model training setup

Models were trained following the setup of Penedo et al. (2024a). We used the GPT-2 tokeniser, Llama architecture and the same training settings: 1.71B parameters (1.82B with embeddings), sequence length of 2048 tokens, and a global batch size of ∼2 million tokens.

Models were trained for 50,000 iterations, which amounts to 100 billion tokens. Training was done on LUMI supercomputer on 16 nodes, each with a single 64-core CPU and 4 MI250x GPUs with dual-GCD (graphics compute die). We used NVIDIA's `Megatron-LM` [4] training framework instead of HuggingFace's `nanotron`[5] framework used by Penedo et al. (2024a).

Training each model took approximately 84 hours, amounting to 10,752 GPU hours per model with the node setup described above, with average performance of 25 teraflops per second. Evaluation was carried out on a single GPU, and with 19 models, each with 50 checkpoints requiring approximately 20 minutes to evaluate, this added 300 GPU hours to the total computational cost. Other processing tasks, such as sampling and tokenisation, required no GPU resources and used a comparatively negligible amount of CPU hours.

## C  Comparison to other data curation schemes

As discussed in Section 2, many different training data selection schemes and tools have been introduced in the literature. To quantify the connection and possible similarities between our register scheme and three other publicly available quality classification tools, FineWeb-edu classifier (Penedo et al., 2024a), the DCLM quality classifier (Li et al., 2024), and NVIDIA's NemoCurator Quality Classifier DeBERTa (He et al., 2023), we classify a sample of over 50 000 documents from our pretraining material with all four classifiers. We compare the selected tool to our register classification scheme by conducting a $\chi^2$-test to evaluate dependence and also report metrics pertaining to the possible overlap.

The FineWeb-edu classifier assigns documents an ordinal label ranging from 0 to 5, the NemoCurator a label 0, 1, or 2, and the DCLM classifier uses a binary label (0,1). For registers, we use the classes in Table 1 under "Individual registers". Obtained results from a $\chi^2$-test (considering FineWeb-edu and NemoCurator's labels nominal) show that all quality classification tools are dependent on the register classification results with $p < 0.01$. This dependence between the classification schemes is to be expected, as all schemes are used as a proxy for "quality" in LLM pretraining. To evaluate the strength of this connection, we calculated Cramer's V, which yielded the following results:

- Registers – DCLM classifier: 0.136
- Registers – FineWeb-edu classifier: 0.221
- Registers – NemoCurator classifier: 0.419

In the context of machine learning, these values indicate a low to moderate connection between the classifiers, with the NemoCurator showing the greatest association, which can be considered somewhat substantial. To account for the two ordinal classification schemes, FineWeb-edu and NemoCurator, we also calculated the $\eta^2$-metric, which measures the amount of impact a variable has on the variance of another. These results were 0.173 and 0.273, for FineWeb-edu and NemoCurator, respectively, which similarly show a strong connection between the schemes but do not imply direct overlap.

Although the above analysis shows associations between the studied schemes, the register classes coincide with all classes of the three tools, with the exception of FineWeb-edu's highest quality class 5, which only occurs in conjunction with register classes *Instructive-Informational* and *Informational Description*. We visualise this in Figures 4, 5, and 6. This means that our register scheme can be seen as mutually reinforcing when combined with other established quality labelling schemes. We specifically see from our results that the 3 tools we compared to the register scheme often assign high-quality labels to documents from registers *Informational Description*, *Description*, and the case of NemoCurator, *News*, but neglect classes that we found beneficial in this study, such as *How-to-Instructions* and *Opinion*. This highlights the value of using the non-binary but linguistically motivated and versatile scheme that the registers offer for curating data.

---

[4]https://github.com/NVIDIA/Megatron-LM
[5]https://github.com/huggingface/nanotron

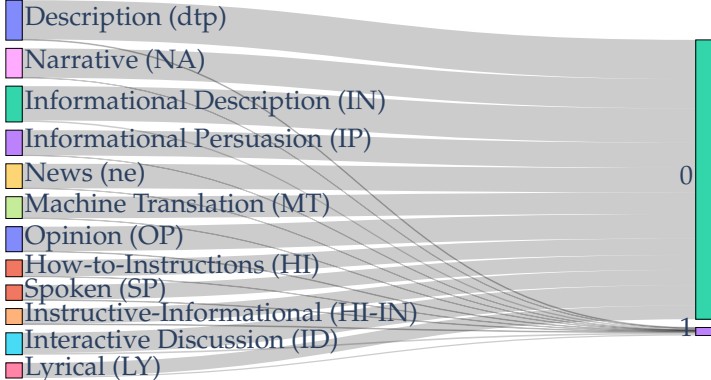

Figure 4: Visualised cross-classification for our register scheme and the DCLM quality classifier. DCLM label 0 corresponds to low quality and label 1 to high quality.

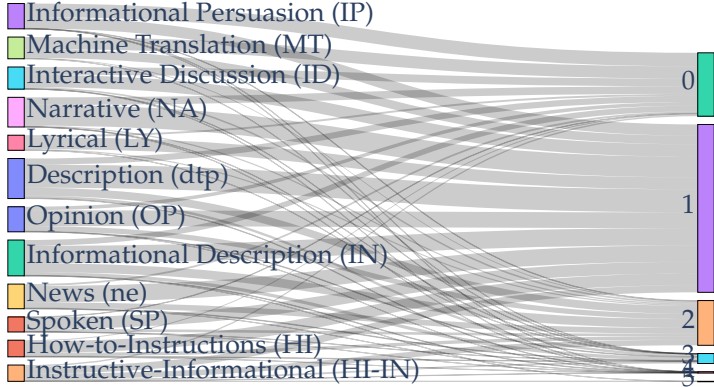

Figure 5: Visualised cross-classification for our register scheme and the FineWeb-edu classifier. FineWeb-edu label 0 corresponds to the lowest quality and label 5 to the highest quality.

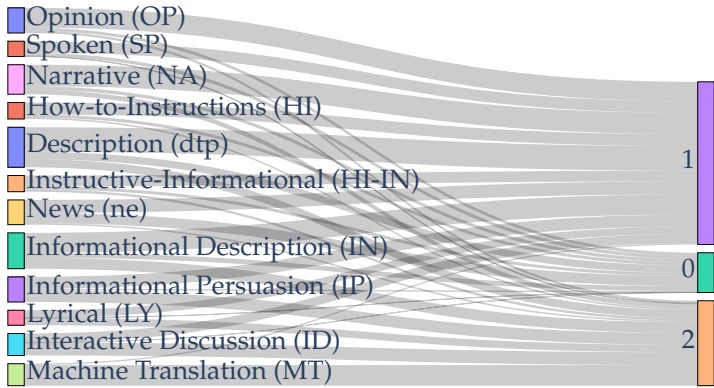

Figure 6: Visualised cross-classification for our register scheme and the NVIDIA NemoCurator classifier. NemoCurator label 0 corresponds to the highest quality and label 2 to the lowest quality.

## D   Numerical results

We present the results in numerical format in Table 3. As stated in Section 3.3, we reweigh the accuracy outputted by Lighteval, acc_norm, as the calculation of acc_norm is heavily biased against MLLU. MMLU consists of 57 tasks, which are considered separate benchmarks by LightEval. This is why values of acc_norm are lower than "Accuracy" in the table: average performance of our models on MMLU is just under 0.3.

| Pretraining dataset | Step | Tokens | Accuracy | acc_norm | stderr |
|---|---|---|---|---|---|
| FineWeb | 1000 | 2 B | 0.348450 | 0.269707 | 0.030520 |
|  | 50000 | 105 B | 0.465567 | 0.330767 | 0.031929 |
| *How-to-Instructions* (HI) | 1000 | 2 B | 0.351006 | 0.264036 | 0.030214 |
|  | 50000 | 105 B | 0.446558 | 0.294467 | 0.030880 |
| *Instructive-Informational* (HI-IN) | 1000 | 2 B | 0.352893 | 0.266001 | 0.030249 |
|  | 50000 | 105 B | 0.464502 | 0.314607 | 0.031506 |
| *HI-IN-HI-dtp* | 1000 | 2 B | 0.354339 | 0.265349 | 0.030291 |
|  | 50000 | 105 B | 0.465616 | 0.320815 | 0.031642 |
| *HI-IN-HI-dtp-OP* | 1000 | 2 B | 0.352104 | 0.267529 | 0.030357 |
|  | 50000 | 105 B | 0.475093 | 0.331857 | 0.032058 |
| *HI-IN-HI-dtp-OP-NA* | 1000 | 2 B | 0.354445 | 0.264463 | 0.030214 |
|  | 50000 | 105 B | 0.472279 | 0.319987 | 0.031603 |
| *HI-IN-HI-dtp-OP-NA-ID* | 1000 | 2 B | 0.348062 | 0.263200 | 0.030155 |
|  | 50000 | 105 B | 0.463591 | 0.324652 | 0.031859 |
| *HI-IN-HI-dtp-OP-NA-ID-SP* | 1000 | 2 B | 0.353098 | 0.268480 | 0.030369 |
|  | 50000 | 105 B | 0.464738 | 0.325085 | 0.031820 |
| HPLT v2 | 1000 | 2 B | 0.341536 | 0.265107 | 0.030274 |
|  | 50000 | 105 B | 0.457484 | 0.328140 | 0.032064 |
| *Interactive Discussion* (ID) | 1000 | 2 B | 0.336598 | 0.264045 | 0.030252 |
|  | 50000 | 105 B | 0.430686 | 0.298429 | 0.031214 |
| *Informational Description* (IN) | 1000 | 2 B | 0.344821 | 0.270357 | 0.030517 |
|  | 50000 | 105 B | 0.437347 | 0.333771 | 0.032217 |
| *Informational Persuasion* (IP) | 1000 | 2 B | 0.336226 | 0.261104 | 0.030146 |
|  | 50000 | 105 B | 0.393026 | 0.276895 | 0.030480 |
| *Lyrical* (LY) | 1000 | 2 B | 0.322654 | 0.261681 | 0.030248 |
|  | 50000 | 105 B | 0.357626 | 0.271479 | 0.030604 |
| *Machine Translation* (MT) | 1000 | 2 B | 0.331829 | 0.262588 | 0.030269 |
|  | 50000 | 105 B | 0.374349 | 0.281125 | 0.030821 |
| *Narrative* (NA) | 1000 | 2 B | 0.336791 | 0.263844 | 0.030236 |
|  | 50000 | 105 B | 0.441469 | 0.297377 | 0.031149 |
| *Opinion* (OP) | 1000 | 2 B | 0.343446 | 0.262086 | 0.030109 |
|  | 50000 | 105 B | 0.446629 | 0.318284 | 0.031720 |
| *Spoken* (SP) | 1000 | 2 B | 0.340778 | 0.266882 | 0.030336 |
|  | 50000 | 105 B | 0.422314 | 0.303859 | 0.031381 |
| *Description* (dtp) | 1000 | 2 B | 0.347940 | 0.268266 | 0.030463 |
|  | 50000 | 105 B | 0.452347 | 0.329816 | 0.032027 |
| *News* (ne) | 1000 | 2 B | 0.332720 | 0.257563 | 0.029931 |
|  | 50000 | 105 B | 0.417933 | 0.304612 | 0.031457 |

Table 3: Accuracies for all models, first and last checkpoint. "Accuracy" denotes accuracy weighted by benchmark, which we present in our figures. Reweighing was done to mitigate the dominance of MMLU in the results, see Section 3.3. acc_norm and stderr stand for average accuracy and standard error as given by LightEval.

## E   Performance by benchmark and register

Figures 7, 8, and 9 present the results over all checkpoints, with Figure 7 showing results by register, others by benchmark. To augment readability, we again applied a rolling average. In Figure 7, the values are normalised with respect to the average benchmark score on the final checkpoint; for example, the *Description* (dtp) model outperforms the average of all registers by 0.1 points on the final checkpoint on the *ARC Easy* benchmark.

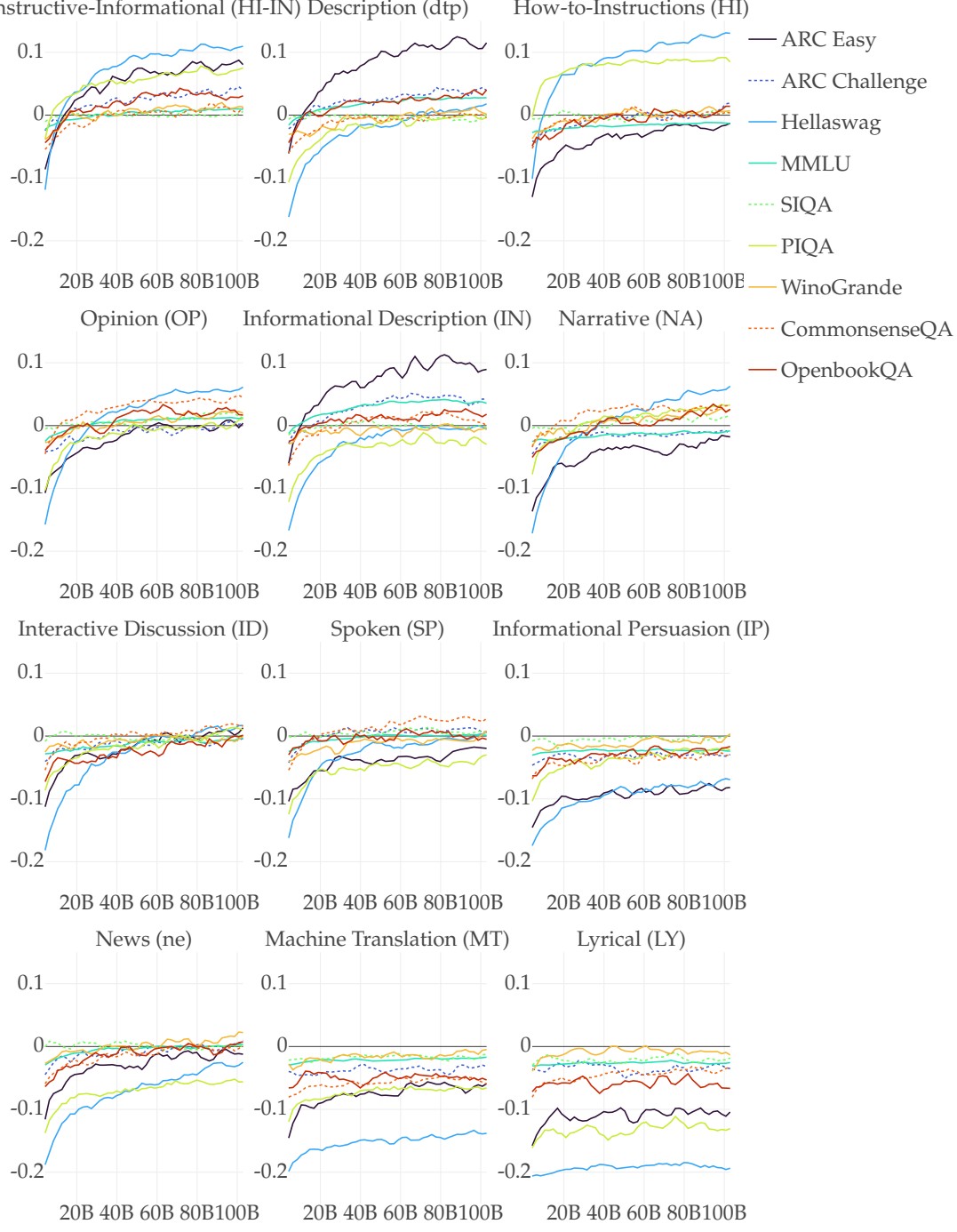

Figure 7: Models' performance by register. Results are scaled by the average last checkpoint performance to highlight the models' differences. Dotted lines to increase readability.

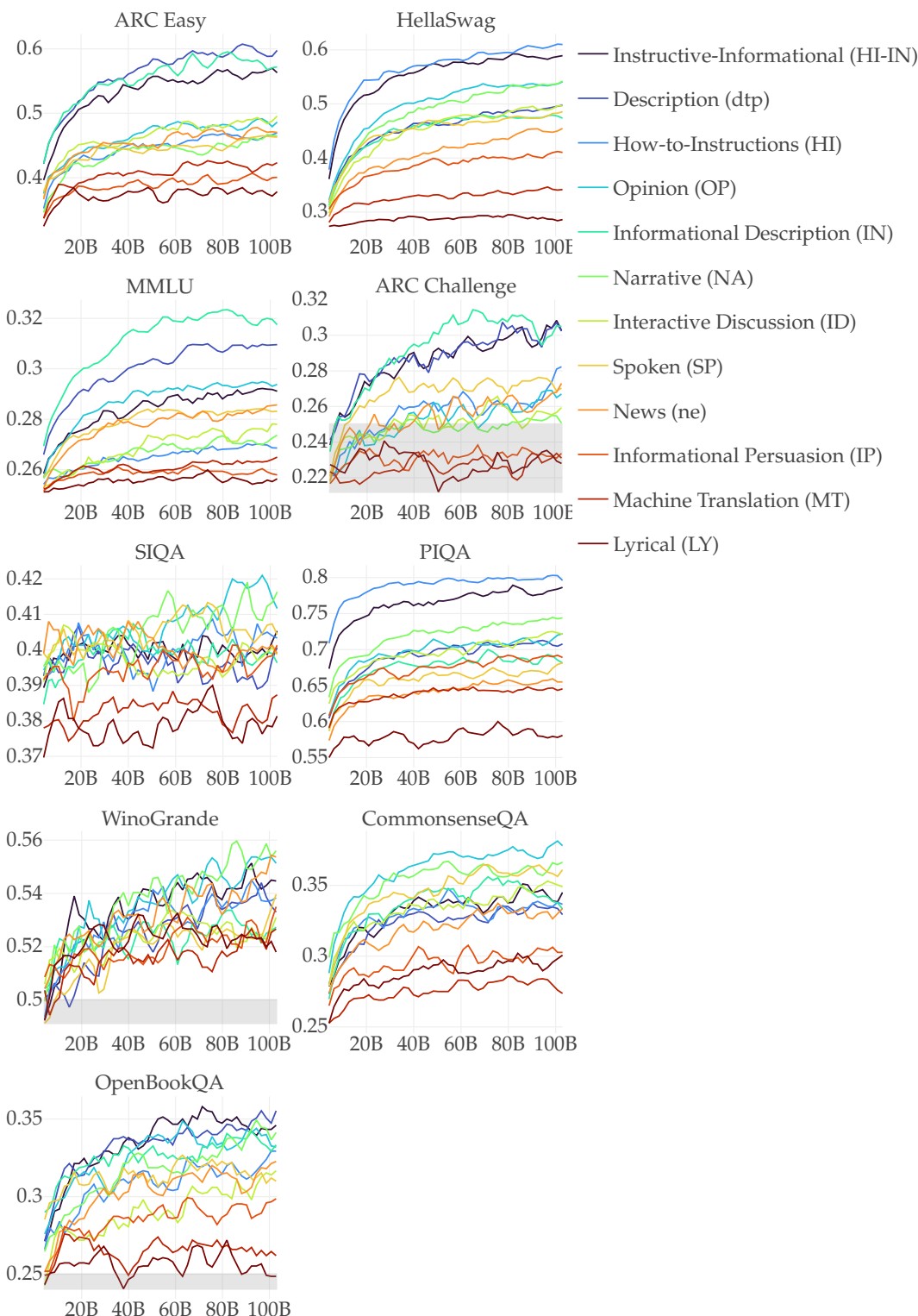

Figure 8: Accuracies of register models on individual benchmarks. Random-guess thresholds shown in grey when at least one model falls under it. Rolling average over 6 billion tokens applied for ease of reading overlapping lines.

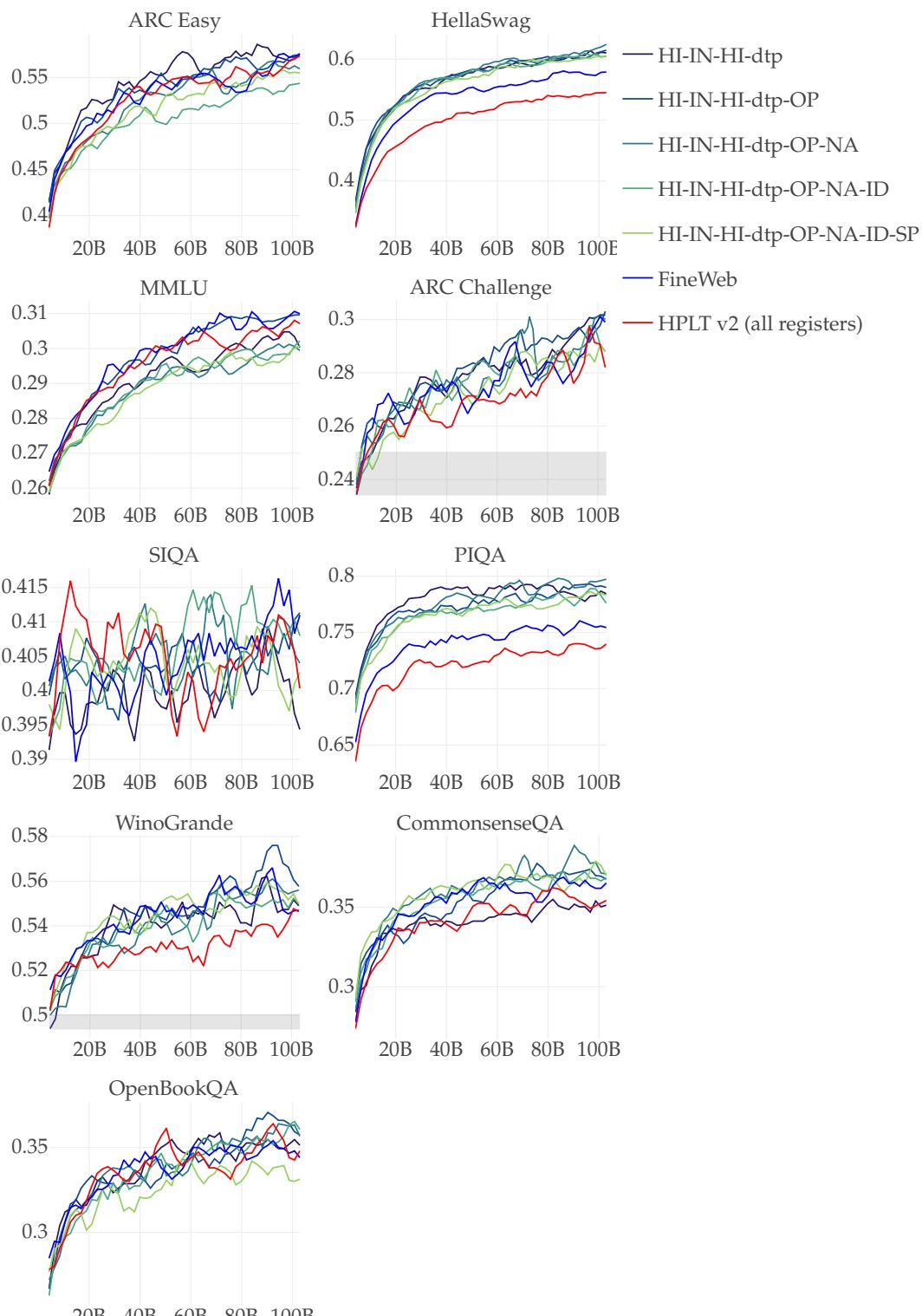

Figure 9: Accuracies of combination models on individual benchmarks. Random-guess thresholds shown in grey when at least one model falls under it. Rolling average over 6 billion tokens applied for ease of reading overlapping lines.

