# OpenReview forum: "Register Always Matters: Analysis of LLM Pretraining Data Through the Lens of Language Variation"
_colmweb.org/COLM/2025/Conference — COLM 2025_

### Official Review · Reviewer_W4AW · 2025-04-28

**Rating:** 7
**Confidence:** 3
**Ethics Flag:** 1

**Summary:**

This paper describes experiments on training LLMs on data from different registers (genres) and how it affects performance on a given selection of tasks. The authors separate the training data into different subsets based on genre/register and train models both on individual subsets and combinations thereof. All models are evaluated on 9 benchmarks (all English).

The paper is well put together and easy to follow, but I'm left a bit unsure as to what the conclusion is: A mix of genres seems better overall, but performance on the individual tasks varies depending on the subset of data used. This seems quite intuitive? Models trained on certain registers perform bad across all tested benchmarks, e.g. lyrical which consists of song lyrics and poems. But the benchmarks all require the model to perform some sort of reasoning, so it really is not surprising that lyrical texts do not work well as training data here?

**Questions To Authors:**

As stated above, the paper is really well written and clear, so I have only one question:

What exactly is the MT register? Is this machine translated text? The English side of parallel corpora used to train MT models? But both of those would consist of different genres/registers (news, instructions, opinion texts, etc.), so I'm not sure how this constitutes its own register? Or is it something else?

**Reasons To Accept:**

The paper is a good read, easy to follow and with good graphics to condense the large number of results.

**Reasons To Reject:**

As stated above, I'm a bit unsure as to what the conclusion or finding is - a lot of it seems quite intuitive:
1. genre/register matters for performance on a given task
2. some combinations perform better than using the entire dataset

Regarding 1., this is not surprising and for 2., how specific are these combinations to the selected benchmarks?

---

> ### Author Response · Authors · 2025-05-30
>
> We thank you for your review and appreciate your recognition of the merits of our work. You bring up important points on our conclusions.
>
> To elaborate on our overarching idea, our goal was to highlight that purely linguistic data selection, which has not been studied before, can yield comparable results to data selection with other methods, with the added bonus of decades of research on the contents of the register classes. We think that our main contribution is that we showed that the previous sentiment is true, and that there are tangible differences in the task-wise performance between these classes, which can be used to inform data selection and to explain model performance. While it is clear that some parts of data lead to better LLM performance, we were able to show that a scheme developed in another field can be useful in LLM training, and we found surprising results, specifically with Opinion class being beneficial and News being detrimental: despite the general finding that register matters is expected, our work goes well beyond establishing this general point experimentally in quantifying not only the overall range of variation in downstream performance (how much register matters) but also the specific impacts of different registers and their combinations for a set of established community-standard benchmarks.
>
> As a first study in the intersection of two deep and broad lines of research -- the linguistic study of web registers and research into LLM training -- we acknowledge that our work leaves a number of questions incompletely answered. We would nevertheless like to argue that our study represents a substantial effort, and openly releasing the many models created in our work makes addressing these and many other questions easier in future work, and consequently we hope that our study is only the first step in exploring the effects of register on LLMs. Releasing the models also allows for further evaluations, by us in the future or by others interested in the effect of registers on LLMs.
>
> Continuing, as you correctly stated, it is no surprise that Lyrical and Machine Translated texts result in poor-performing models. However, we think that running the experiment with all main-level registers, which together cover the full linguistic variation of the data, instead of predetermined subsets, was the most honest and transparent way to do this analysis. Although we additionally chose to include the “news (ne)” and “description of a thing or a person (dpt)” categories, this was done because there has been specific interest in using news articles and Wiki-pages as LLM training data.
>
> While we do agree with you second point, benchmark selection is a broad question that is arguably relevant to all studies exploring LLM performance. This does not mean we think this criticism is not valid, or that this makes us exempt from it. However, our goal with benchmark selection was to maximise comparability by using benchmarks that have already been used to evaluate models in previous ablation studies, and using benchmarks that are established in the field, which is why we decided on the FineWeb tasks. Penedo et al. [1] also have already ascertained that these benchmarks are suitable for evaluating small models, which we think is an important part of the evaluation setup to produce reliable and generalisable results.
>
> Finally, we would like to answer your question about machine translation: MT is a separate class in the register classification scheme, meaning the register classification model has been trained to recognize text with linguistic features commonly seen in machine translation. This is specifically implemented because machine translation has its own features, which are distinct from other registers. This means that the MT category can contain documents which were originally some other register, but the machine translation has removed some of these features, and thus it is more justifiable to have it as its own category.
>
> We thank you again for the good questions that will help us make our work better. Specifically, answering these questions has allowed us to find many places in the article that require a better explanation, and we are more than happy to include those. We hope you continue the discussion with us!
>
>
> [1] Penedo et al. (2024) The FineWeb Datasets: Decanting the Web for the Finest Text Data at Scale, https://proceedings.neurips.cc/paper_files/paper/2024/hash/370df50ccfdf8bde18f8f9c2d9151bda-Abstract-Datasets_and_Benchmarks_Track.html

---

> ### Comment · Reviewer_W4AW · 2025-06-06
>
> Thank you for taking the time to compose such a detailed reply! I understand the motivation behind your experiments better after reading your explanations, and I will up my score to reflect that (maybe you could include some of this in the introduction).

---

> > ### Author Response · Authors · 2025-06-06
> >
> > Thank you for taking the time to read our reply and for updating the score! We will add some of the discussed points to our paper to better reflect our motivation for these experiments.

---

### Official Review · Reviewer_syA5 · 2025-05-12

**Rating:** 2
**Confidence:** 4
**Ethics Flag:** 1

**Summary:**

This article reports an interesting observation: it is possible to train an LLM on a smaller corpus, provided that the data is filtered to include only certain linguistic registers, without significantly affecting its performance on standard benchmarks.

It joins a long line of research on corpus filtering training, to which it makes a relevant contribution.

**Reasons To Accept:**

A well-written, interesting article that reports an interesting observation that might be useful for people who want to train LLMs.

**Reasons To Reject:**

This article raises several concerns that, in my opinion, currently prevent it from being suitable for publication, despite the fact that I find the main conclusion interesting.

First, regarding matters of form: I believe it is essential to provide a detailed description of the model architecture in the main text of the paper. This is not a minor detail that can be relegated to an appendix — it is central to understanding and evaluating the experimental setup.

Second, I do not understand the rationale behind presenting evaluation results across all training checkpoints rather than focusing on final performance (e.g., the last or best checkpoint). The evaluation strategy is never explicitly discussed by the authors, and this choice makes it particularly difficult to interpret or compare results — especially between Figures 1 and 2, which are central to the article. A table summarizing final metrics would have enabled a clearer and more quantitative comparison.

It would also have been valuable to include a comparison of training costs across models. This would help assess to what extent reducing the training corpus size actually translates into practical efficiency gains.

More fundamentally, the article feels incomplete: the authors report an intriguing observation — that some corpora yield better-performing LLMs — but make no attempt to explain it. While it may seem trivial that a poorly constructed dataset would lead to underperformance, and while it is well known that training and evaluation distributions should align, it would have been insightful to explore which properties of the data might influence model quality. Addressing this would also contribute to a more foundational question: how can we select training data without having to train models on all possible registers or combinations thereof? As it stands, the work merely shows that such a selection might be beneficial, without providing any guidance on how to make it. A straightforward experiment, which could be easily conducted, would help begin addressing my concerns: it would be informative to examine how the classifier used for data filtering categorizes the various examples in the test corpus.

Finally, a broader question: would it not have been relevant to investigate the impact of the model architecture — especially its size? Could reducing the training data also be a strategy for justifying or enabling smaller models?

---

> ### Author Response · Authors · 2025-05-30
>
> We thank you for your review and appreciate your recognition of the merits of our work. While we were disappointed to see what we consider a harsh overall rating, we are happy to note that we can fully address the given reason to reject, as detailed below.
>
> > It is essential to provide a detailed description of the model architecture in the main text of the paper
>
> We have now moved the detailed description from the appendix into the main text of the paper, which hopefully fully addresses this reason to reject. We would like to note that in the submitted version, we identified the model size and architecture in the main body of the text and the completed details in an appendix, which follows prominent recent work in this area (e.g. [1,2,3]). There thus appears to be some divergence in views on whether this practice constitutes a reason to reject a manuscript.
>
> > I do not understand the rationale behind presenting evaluation results across all training checkpoints rather than focusing on final performance
>
> We have moved the numerical evaluation results for the final checkpoint performance into the main body of the manuscript. We note that these results are consistent with the current results and do not impact the conclusions of our work. We have also updated the draft to identify some of the advantages of analyzing performance across all training checkpoints, including 1) insight into consistent trends seen across the training process and 2) as a proxy for estimating performance across different training budgets in terms of the number of tokens in settings where training separate models is prohibitively expensive.
>
> > It would also have been valuable to include a comparison of training costs across models.
>
> Thank you for the suggestion, we have added the training costs of the models into the draft.
>
> > would it not have been relevant to investigate the impact of the model architecture — especially its size?
>
> These are broad questions that are arguably relevant to all studies exploring training data. While we agree that exploration of different model architectures and scales would generally strengthen such work – including ours – we would like to note that such experiments increase both the effort and computational cost, and prominent studies focusing on training data focus on single established architectures and model scales (e.g. [1,2,3]), while the impact of model size has been studied by [4,5,6,7]. While we acknowledge that our study leaves questions such as these open, we would like to argue that our study represents a substantial effort for a conference paper and is in line with the broader community practice.
>
> > [...] a more foundational question: how can we select training data without having to train models on all possible registers or combinations thereof?
>
> Data ablation studies are done since there is currently no known method of selecting subsets of data, registers in our case, that are beneficial for model training without trying different combinations. The results of our experiments may pave the way for future, more general research on the effect of register on LLM training.
>
> > [...] it would be informative to examine how the classifier used for data filtering categorizes the various examples in the test corpus.
>
> Register classification and the contents of each register class have been studied by [8,9,10] among others. We chose the linguistic register specifically because of the large amount of linguistic and machine learning research into the contents of the classes. We realize this perspective could be stated more clearly in the paper, and have updated the manuscript to reflect this. Alternatively, in case you meant running the classifier on the benchmarks to analyze their contents and linguistic characteristics and their correlation with the register’s performance, this would provide an interesting avenue for future research.
>
> We again thank you for the review, hope we were able to convince you of our work and that you continue the discussion with us!
>
>
> [1] Soldaini et al. (2024) Dolma: an Open Corpus of Three Trillion Tokens for Language Model Pretraining Research
>
> [2] Li et al. (2024) DataComp-LM: In search of the next generation of training sets for language models
>
> [3] Penedo et al. (2024) The FineWeb Datasets: Decanting the Web for the Finest Text Data at Scale
>
> [4] Kaplan et al. (2020) Scaling Laws for Neural Language Models,
>
> [5] Roberts et al (2020), How Much Knowledge Can You Pack Into the Parameters of a Language Model?
>
> [6] Hoffman et al (2022), Training Compute-Optimal Large Language Models
>
> [7] Hassied et al. 2024, The Larger the Better? Improved LLM Code-Generation via Budget Reallocation
>
> [8] Biber, D. and Egbert, J. (2018). Register Variation Online.
>
> [9] Kuzman and Ljubešic (2023). Automatic genre identification: A survey.
>
> [10] Laippala et al. (2021). Exploring the role of lexis and grammar for the stable identification of register in an unrestricted corpus of web documents

---

> > ### Author Response · Authors · 2025-06-10
> >
> > Thank you again for your review. We would appreciate your comment on our response, as we think we were able to address your concerns.

---

> > > ### Comment · Reviewer_syA5 · 2025-06-11
> > >
> > > Thank you to the authors for their responses. However, I believe that the suggested revisions are substantial, and as such, it is not possible to accept the paper based on the discussion alone without a full evaluation of the revised manuscript. Moreover, my main concern – namely, that the observation is rather trivial and the authors do not discuss how their findings could be applied in practice without training a multitude of models – has not been properly addressed.

---

> > > > ### Author Response · Authors · 2025-06-11
> > > >
> > > > Thank you for your response! It is unfortunate that there is so little time left, but we would appreciate the opportunity to continue this discussion.
> > > >
> > > > > I believe that the suggested revisions are substantial, and as such, it is not possible to accept the paper based on the discussion alone without a full evaluation of the revised manuscript.
> > > >
> > > > The revisions we suggested primarily concern moving material already present in the submitted manuscript from the appendix to the main body of the paper and adding the training cost, which is trivial to estimate using the conventional 6ND approximation (where N is number of parameters and D the number of training tokens) and the throughput, which averages approximately 140 TFLOPs/GPU in our case. We respectfully disagree with the characterization of these revisions as substantial and would hope that we could be relied on for performing such editing correctly.
> > > >
> > > > > [...] the observation is rather trivial, [...]
> > > >
> > > > We regret that you now perceive our observation of register effects on LLM performance as trivial, compared to your original review in which you characterized our work in part as follows: "This article reports an interesting observation"; "[...] article that reports an interesting observation"; "the authors report an intriguing observation". Respectfully, the reason we did not address criticism that the observation is trivial is that we did not read the original review as implying that this was a main concern, and appreciate the opportunity to respond to this now:
> > > >
> > > > Our goal was to highlight that data selection based purely on linguistic register, which has not been studied before in the context of LLM training, can yield comparable results to data selection with other methods, with the added bonus of decades of research on the contents of the register classes, that can be used to explain their effects on the models. Our work goes well beyond establishing this general point experimentally in quantifying not only the overall range of variation in downstream performance (how much register matters) but also the specific impacts of different registers and their combinations for a set of established community-standard benchmarks.
> > > >
> > > > Most importantly, we would argue that contrary to not "explor[ing] which properties of the data might influence model quality" the core of our work is precisely this exploration -- we anticipated that linguistic register (a property of the data) would influence model quality, and not only demonstrated such an effect overall but also detailed it in further experiments on register combinations. To the best of our knowledge, this is the first study to explore the impact of register, a well-studied characteristic of data in linguistics, on model quality.
> > > >
> > > > > [...] authors do not discuss how their findings could be applied in practice without training a multitude of models [...]
> > > >
> > > > There may have been a misunderstanding of your original comment here. We assumed that you were criticizing the common practice of performing ablation studies to select and evaluate subsets of data, but perhaps you are instead referring to how to apply the findings resulting from these experiments. Here, the answer is simply that register labels found in datasets such as HPLT and readily assigned to other datasets using lightweight classifiers introduced in previous work can be directly used to select which subsets of the data to use, as demonstrated in our work. We point this out in the original manuscript in the abstract on rows 27-29 and rows 349-357:
> > > >
> > > > __"These findings show that register is an important explainer of model variation and can facilitate more deliberate future data selection practices."
> > > > "Our findings show that register is an important explainer of LLM performance,[...] These findings can be used to increase pretraining data transparency and to optimise data selection methods, potentially addressing specific performance gaps by incorporating appropriate registers."__
> > > >
> > > > The way to apply our findings in practice is thus identical to the way in which the data quality analysis results of studies such as FineWeb-edu, DCLM, and Nemotron-CC can be applied in practice. We might even argue that our results have potentially broader applicability as register labels are available for many languages other than English.
> > > >
> > > > Thank you for your answer, and we hope that there is still time to continue the discussion, as we firmly believe we have answered your concerns!

---

### Official Review · Reviewer_ak3C · 2025-05-13

**Rating:** 7
**Confidence:** 4
**Ethics Flag:** 1

**Summary:**

This paper studies the effect of pretraining data curation by register on language model performance. Register refers to different genres of text, such as how-to-instructions, narrative, and lyrical text. The paper performs a large number of data ablation studies by pretraining many 1.7b-parameter Llama models on different registers and on combinations of registers. The suite of trained models is evaluated on multiple general-purpose downstream tasks, such as HellaSwag and MMLU. The paper finds that some registers, like instructive-informational, lead to higher performance when trained on in isolation, and that combinations of registers lead to even higher performance.

**Questions To Authors:**

1. In Table 1, you list descriptions for the combinations as, e.g. '1/3rd of HI-IN, HI, dtp'. What does this mean? The number of available tokens for the combinations don't match up with 1/3rd of each of the individual registers.
2. Your conclusion mentions that some relationships between register and downstream performance. Could you elaborate on this?

**Reasons To Accept:**

1. The paper performs a large data curation ablation study that can help model pretrainers decide how to curate data. The experimental setup is clear and sound, with appropriate baselines and care taken to account for different dataset sizes.
2. The paper offers clear findings, such as training on instructive-informational data leads to higher downstream performance on the evaluated tasks than training on other registers.

**Reasons To Reject:**

1. The paper would be made stronger by situating your results against those of other pretraining data curation papers that train on different kinds of data. While this is the first paper to my knowledge that curates based on register, some of the papers you mention in related work do curate based on similar notions of topic distribution or data source. For example, how do your findings complement the data domain ablations in Figure 5 in Longpre et al. 2024 or the coverage experiments in Sachdeva et al. 2024?
2. You evaluate all of your models on general-purpose benchmarks like HellaSwag and MMLU. What about more targeted evaluations that might highlight strengths and weaknesses of different registers? I know you mention this in the limitations section already, but it does seem relevant to determining the potential benefits of different registers.

---

> ### Author Response · Authors · 2025-05-30
>
> We appreciate your recognition of the merits of our work and thank you for your review which brings up important questions about our work.
>
> It is correct that the overlap between our classification system and those used in literature needs to be studied. Following your suggestion, we conducted an additional experiment by using the FineWeb-edu-classifier [1] and the DCLM quality classifier [2], which were readily available on Huggingface, unlike the classifiers used in Longpre et al. 2024 (same as in GLAM by Du et al. 2022) and Sachdeva et al. 2024, which were suggested. We labelled a large number of documents with all classifiers and found that our register classification does not correlate substantially with either quality filtering method, and thus our register scheme does add a new perspective on dataset curation. We have now included these results and further statistical comparisons in the draft, and we hope this addresses one of the reasons to reject.
>
> The results of this experiment also showed that while the FineWeb-edu-classifier frequently gave high scores to news, our results show that news texts lead to weak models. We think that this is a key difference in the register classification, as it focuses on the language, and not the topics of the documents, and which makes it different from many other approaches.
>
> You also mention the general domain benchmarks we use. While we do agree, benchmark selection is a broad question that is arguably relevant to all studies exploring LLM performance. This does not mean we do not think this criticism is not valid, or that this makes us exempt from it. However, our goal with benchmark selection was to maximise comparability by using benchmarks that have already been used to evaluate models in previous ablation studies, and using benchmarks that are established in the field, which is why we decided on the FineWeb tasks. Penedo et al. [1] also have already ascertained that these benchmarks are suitable for evaluating small models, which we think is an important part of the evaluation setup to produce reliable and generalisable results.
>
> As a first study in the intersection of two deep and broad lines of research -- the linguistic study of web registers and research into LLM training -- we acknowledge that our work leaves a number of questions incompletely answered. We would nevertheless like to argue that our study represents a substantial effort, and openly releasing the many models created in our work makes addressing these and many other questions easier in future work, and we hope that our study is only the first step in exploring the effects of register on LLMs. Releasing the models also allows for further evaluations, by us in the future or by others interested in the effect of registers on LLMs.
>
> Lastly, answers to your questions:
>
> 1. ⅓ HI-IN-HI-dpt means we sample uniformly from classes “HI-IN”, “HI”, and “dtp”, using 33.3B tokens from each class. The Available-column indicates that if we were to continue training with these fractions, our cutoff point would be 210B tokens, as then a full epoch of HI-IN (70B tokens in total) is used. We agree this part needs further clarification and we have now added an explanation.
>
> 2. In the conclusion by “downstream performance” we mean the models’ performance on the selected benchmarks. This is to highlight the conclusion that the linguistic register has an effect on the models when evaluated on different tasks. This choice of word is not the best, and we have corrected this to be clearer.
>
> Thank you again for the review, and we hope we have addressed your concerns, and we hope you continue the discussion with us!
>
> [1] Penedo et al. (2024) The FineWeb Datasets: Decanting the Web for the Finest Text Data at Scale, https://proceedings.neurips.cc/paper_files/paper/2024/hash/370df50ccfdf8bde18f8f9c2d9151bda-Abstract-Datasets_and_Benchmarks_Track.html
>
> [2] Li et al. (2024) DataComp-LM: In search of the next generation of training sets for language models
> https://proceedings.neurips.cc/paper_files/paper/2024/hash/19e4ea30dded58259665db375885e412-Abstract-Datasets_and_Benchmarks_Track.html

---

> > ### Comment · Reviewer_ak3C · 2025-06-06
> >
> > Thanks for your detailed response!
> >
> > I appreciate the comparison between linguistic registers and the quality filters of FineWeb-edu-classifier and the DCLM classifier. The orthogonality of registers and existing quality filters improves the paper. I'm looking forward to seeing the full results in the final paper.
> >
> > Could you comment on the relation to existing papers that ablate data from specific sources? This is what I was referring to when I mentioned the data domain experiments in Figure 5 of Longpre et al, not their quality filtering experiments. Documents in the Pile are annotated by their source, and they leave out one source domain (e.g. social media data or PubMed data) at a time. Appendix M of the Dolma paper [1] also experiments with different data mixes based on domain metadata (their Table 4 and Figure 12). It seems to me that your register-specific classifications are more fine-grained than source domain meta-data for web-scraped documents, but I think it would be useful to highlight this in the paper. Do you have any data on how the registers cut across or conform to source domain meta-data?
> >
> > And thanks for answering my clarification questions. My second question was accidentally missing a word: I meant to ask about the 'surprising' results. But I'm satisfied by your discussion on this topic with Reviewer W4AW.
> >
> > [1] Soldaini et al, 2024. Dolma: an Open Corpus of Three Trillion Tokens for Language Model Pretraining Research. https://aclanthology.org/2024.acl-long.840.pdf

---

> > > ### Author Response · Authors · 2025-06-09
> > >
> > > Thank you for your response!
> > >
> > > The HPLT dataset we use offers analytics (https://dattest.prompsit.com/viewer), including URL distribution, but these results seem to be unfortunately missing for English. Other than the pure source URL, the HPLT dataset does not include other domain metadata. This means we cannot answer your question quantitatively without major new experiments, but we will comment qualitatively:
> > >
> > > The connection between web register/genre and the domain of documents has been considered as automatic register identification has developed over the past years: Before the emergence of powerful deep learning methods for NLP, register classification using URLs, HTML tags, etc., was investigated [1, 2]. Modern approaches that do not rely on URLs outperform these methods substantially [3, 4, 5]. Thus, from our perspective, strictly aligning documents' source URLs and registers has proven to be an unfruitful direction to use on its own. Our register scheme was specifically designed to target the entire web, to be adaptable for multiple purposes and end applications; of course, some registers will overlap mostly with certain domains, but as you correctly stated, our register scheme offers improvement by the means of granularity: using the full scheme with over 20 labels, we can sort everything from the domain "nytimes.com" to multiple categories: news, sports reports, op-eds, comment sections, etc. We can also do the same thing for any arbitrary news site without needing the explicit knowledge that a domain is a news site, and additionally, this sentiment also applies to other languages, as our register framework has shown multilingual capabilities. We obviously are not using 20+ registers yet, but using the full scheme is one of our end goals for future work, with this experiment being the first step towards that.
> > >
> > > We will include the points above in the paper. Lastly, thank you for taking the time to read our response to another reviewer. We're glad it clarified our motivation!
> > >
> > > [1] Biber D, Egbert J. Register Variation Online. Cambridge University Press; 2018.
> > > [2] Kuzman T, Mozetič I, Ljubešić N. Automatic Genre Identification for Robust Enrichment of Massive Text Collections: Investigation of Classification Methods in the Era of Large Language Models. Machine Learning and Knowledge Extraction. 2023; 5(3):1149-1175. https://doi.org/10.3390/make5030059
> > > [3] Laippala, V., Rönnqvist, S., Oinonen, M., Kyröläinen, A., Salmela, A., Biber, D., Egbert, J., and Pyysalo, S. (2023). Register identification from the unrestricted open web using the corpus of online registers of English. Language Resources and Evaluation, 57(3):1045–1079.
> > > [4] Kuzman, Taja & Ljubešić, Nikola. (2023). Automatic genre identification: a survey. Language Resources and Evaluation. 59. 537-570. 10.1007/s10579-023-09695-8.
> > > [5] Henriksson et al. (2024). Automatic register identification for the open web using multilingual deep learning, https://arxiv.org/abs/2406.19892

---

> > > > ### Author Response · Authors · 2025-06-10
> > > >
> > > > We hope that our responses have adequately addressed your concerns and would like to ask whether you might be willing to consider increasing your score in light of these answers. If you have any remaining concerns, please do let us know!

---

> > > > > ### Comment · Reviewer_ak3C · 2025-06-11
> > > > >
> > > > > Thanks for answering my questions. I'm satisfied with the answers and have updated my score.

---

### Official Review · Reviewer_4S5f · 2025-05-16

**Rating:** 6
**Confidence:** 3
**Ethics Flag:** 1

**Summary:**

This paper presents how documents of different registers in the pre-training data may impact the performance of the trained models. Register is a linguistic concept for the variety of language used for a particular purpose or particular communicative situation. This paper obtains register labels for documents in a large-scale pre-training data source HPLT v2 using a BERT-based classification model, and trains multiple 1.7B models on datasets of different register mixtures. Via evaluating the trained model on multiple standard LLM evaluation benchmarks, the authors observe varied LM performance across different register mixtures, and find certain register mixture can lead to better performance compared to the original document mix.

**Questions To Authors:**

- First, can you provide examples of documents for each of the registers in the dataset? In addition, it seems the categories in table 1 is not the full set of register labels produced by the model (see ErikHenriksson 2024) -- what happens for those removed categories? are they removed?
- I think one important missing category seems to be "code (or programs)" -- existing literature shows that code has a significant impact for language models during pre-training. None of the categories seem suitable for code (that's the reason why I ask about the first question), and it seems in this paper it doesn't talk too much about code. It would be helpful if the authors can provide details for this.
- Further, can the register labeling mixture strategy results in very different mixtures in other datasets? for example, in the dolma paper (dolama: an Open Corpus of Three Trillion Tokens for Language Model Pretraining Research) in appendix M & N, they report detailed ablation results comparing how different data mixtures impact the training. I feel doing the exact same replication studies might be challenging, but some analysis along that direction should be helpful.

Also some questions about figures:
- The visualization of the training is nice -- for figure 3, can you have a different version where you can plot different benchmark results for the same data mixture in one plot, and it would be interesting to compare the trend across different mixtures?
- In figure 4, why there are dotted lines?

**Reasons To Accept:**

- This paper addresses an important problem in training LMs for better understanding the pretraining data and how it impacts the downstream performance. It is clearly written, with proper details for the experimental setup and training/evaluation seems to be reasonable.
- The analysis (and visualization of the results) is nice, and it may offer a new angle to study the pre-training data.

**Reasons To Reject:**

- I have some questions in terms of experimental details and how it is generalizable. I think one key question I have is that to what extent the register mixture is different from existing data selection and mixing methods, or whether ablating registers offer a very different angle for people to analyze the training data. Right now it seems the paper doesn't compare with other data mixing methods, which could be further strengthened with additional details and comparison. (See more in the question section below.) I am happy to increase the scores if the main concerns are addressed.

---

> ### Author Response · Authors · 2025-05-30
>
> We thank the reviewer for insightful comments and appreciate your recognition of the merits of our work. You bring up an important point of comparison with other labelling schemes, it is true that this type of analysis is missing and would be valuable for the paper. This prompted us to run a comparison between our scheme, the FineWeb-edu-classifier, and the DCLM quality classifier, which were used to select data for two state-of-the-art datasets, [1] and [2], respectively.
>
> We ran this experiment by labelling a large number of documents with all of the classifiers. Our results show that our register classification scheme does not correlate substantially with the other two quality classifiers, thus, our scheme adds a new perspective on dataset curation. We further found out that while our results show that “news” leads to suboptimal performance, the FineWeb-edu classifier very often ranks “news” as educational. We think that this is a key difference in the register classification, which focuses on the language, and not the topics of the documents, which makes register valuable and different from many other approaches.
>
> We have now included the above analysis, results and further statistical comparisons in the manuscript. To answer the rest of your questions,
>
> 1.	We will include some example texts to better highlight the contents of the register categories. The difference between the register categories mentioned in Henriksson et al. (2024) and in our work is due to the register classification scheme being hierarchical; we only use the the first level classifications, and additionally chose to include the “news (ne)” and “description of a thing or a person (dpt)” categories, as there has been specific interest in using news articles and Wiki-pages as LLM training data. This means the unmentioned sub-categories are present in the data, we just do not use them to separate classes, so they are omitted in the notation. In hindsight, we see that this could have been explained more clearly and have now corrected this issue.
>
> 2.	As the register focuses on linguistic features, you are correct that none of the categories fit code on the surface. Documents containing code can be found in multiple register categories, as they are categorised by their surrounding context: if the document is instructive, it will be categorized to HI. If it is a coding forum, it will be in ID. Our data source HPLTv2 does not specifically include code-sources, like the Stack, so we relied on the context surrounding the code snippets. Analysing pure code through the lens of our work is an interesting topic for future research.
>
> 3.	We agree this analysis would have strengthened the manuscript, but due to the associated computational cost and effort of evaluating a multitude of register combinations and proportions of mixes we must leave this study for future work. As there are 9 main-level registers, combinations with different mixes would result in hundreds of trainable models. Already, training the 19 models used in the experiments required substantial computational resources.
>
> 4.	Your suggestion to have additional plots by data mix, showing all benchmarks, would lead to nice comparison and is something we will include in the appendices.
>
> 5.	The dotted lines indicate training over one epoch; this information was missing and we have included this in the captions and in the section where we discuss training over one epoch.
>
> Thank you again for the review and helpful suggestions, and we hope we have addressed your concerns. We hope you continue the discussion with us!
>
> [1] Penedo et al. (2024) The FineWeb Datasets: Decanting the Web for the Finest Text Data at Scale, https://proceedings.neurips.cc/paper_files/paper/2024/hash/370df50ccfdf8bde18f8f9c2d9151bda-Abstract-Datasets_and_Benchmarks_Track.html
>
> [2] Li et al. (2024) DataComp-LM: In search of the next generation of training sets for language models, https://proceedings.neurips.cc/paper_files/paper/2024/hash/19e4ea30dded58259665db375885e412-Abstract-Datasets_and_Benchmarks_Track.html

---

> > ### Comment · Reviewer_4S5f · 2025-06-09
> > **Thanks for your response!**
> >
> > Thanks for your response!
> >
> > > We thank the reviewer for insightful comments and appreciate your recognition of the merits of our work. You bring up an important point of comparison with other labelling schemes, it is true that this type of analysis is missing and would be valuable for the paper. This prompted us to run a comparison between our scheme, the FineWeb-edu-classifier, and the DCLM quality classifier, which were used to select data for two state-of-the-art datasets, [1] and [2], respectively.
> > We ran this experiment by labelling a large number of documents with all of the classifiers. Our results show that our register classification scheme does not correlate substantially with the other two quality classifiers, thus, our scheme adds a new perspective on dataset curation. We further found out that while our results show that “news” leads to suboptimal performance, the FineWeb-edu classifier very often ranks “news” as educational. We think that this is a key difference in the register classification, which focuses on the language, and not the topics of the documents, which makes register valuable and different from many other approaches.
> > We have now included the above analysis, results and further statistical comparisons in the manuscript
> >
> > Thanks! I don't think COLM allows updating the PDF -- can you share the detailed results in the comment?

---

> > > ### Author Response · Authors · 2025-06-09
> > >
> > > Thank you for your response. As we are not allowed to update the PDF during the discussion period, we present the results here:
> > >
> > > The FineWeb-edu classifier uses ordinal labels 1-5, the DCLM classifier is binary (0,1), and our register scheme has 12 labels (for this experiment). All values were calculated from 52k samples with the data sampled following the register distribution observed in the HPLT2 dataset. First, we ran a chi^2 test for both experiments (considering FineWeb-edu labels as nominal for this step), which showed that there is an association between the schemes with p-value < 0.01. This is to be expected, as we would hope there is something universal captured by schemes used for quality classification. However, to show that register does not overlap perfectly with existing schemes, making it redundant, we evaluated the strength of this association with Cramer's V metric. Results are
> > > - Cramer's V (registers, FineWeb-edu): 0.21
> > > - Cramer's V (register, DCLM): 0.13
> > >
> > > Both of these imply that although significant, the connection between these schemes is weak.
> > >
> > > However, after posting our first comment, which we based on the above results, we also calculated another metric for the FineWeb-edu label scheme, as we previously considered it nominal despite it being ordinal. We calculated eta^2
> > > - Eta^2 (register, ordinal FineWeb-edu): 0.16
> > >
> > > which does signify an effect; 16% of the variance in FineWeb-edu score can be explained by the register label. While eta^2 of 16% is generally considered large, in our case of machine learning prediction, 16% of variance explained by the register is not large enough to make the register classification one-to-one with the FineWeb-edu classifier, but it does signify a stronger connection.
> > >
> > > The above discussion shows that while there is some expected level of connection between these schemes and registers, the connection is not strong enough to make our approach redundant, and thus, register opens a new window for dataset curation. We also further wanted to show the connections between the schemes using sankey-diagrams (https://imgur.com/a/rK0oB1c, anonymous sharing) for the compared classifiers: from these figures, it can be seen that all registers contain documents predicted as good and bad quality by the DCLM classifier. Similarly, all FineWeb-edu levels contain multiple registers, except for level 5, which in our sample only maps to "IN" register. As stated, we think this is among the strengths of register labels, which focus on different aspects of documents. We will update the manuscript to reflect these findings.
> > >
> > > Thank you again for taking the time to respond to us, and please let us know if there is something else we can further clarify!

---

> > > > ### Author Response · Authors · 2025-06-10
> > > >
> > > > We hope that our responses have adequately addressed your concerns and would like to ask whether you might be willing to consider increasing your score in light of these answers. If you have any remaining concerns, please do let us know!

---

### Author Response · Authors · 2025-06-05
**Request for discussion with reviewers**

We would first like to thank the reviewers again for their many pointers on how to strengthen our work! As we pointed out in our responses, we believe that we have addressed many of the given reasons to reject, and would appreciate the opportunity to discuss with the reviewers to understand whether our responses are satisfactory or if there is anything further we can do to address their concerns.

---

### Comment · Area_Chair_d3xm · 2025-06-05
**Engagement!**

Hey reviewers -- thank you for your time writing the reviews. The authors have put in a solid effort in responding to comments and I encourage you to take a look. The most useful things to consider are on pivot points that could motivate you to change your recommendation for the paper. Let me know if you need anything from the AC level / program committee!

---

### Decision · Program_Chairs · 2025-07-08

**Decision:**

Accept

**Comment:**

This is a cool paper! I'm fairly familiar with pretraining data practices, so I can see how this overlaps with some related work, but I agree with the authors that most of the notion of "register" in modern language modeling is not know. The paper is well written, the experiments make sense, and it is bringing a classic linguistics idea into relatively modern pretraining experiments (~2B models, 100B tokens).

3 of the 4 reviewers agree this is a strong paper and the authors have worked on the manuscript to address their concerns. The one low review score seems to be focusing on more minor issues, i.e. the points raised by reviewer syA5 seem minor detractions to the COLM reviewer / score guidelines where overall the paper checks many boxes

I'm not sure if the method will catch on as being "central" to language modeling development, but I see it being adopted in complement to "domain" sampling that's done today.

Regardless, I am happy to recommend accepting this paper. This seems like a very solid COLM contribution.